# Estimating internationally imported cases during the early COVID-19 pandemic

Tigist F. Menkir[1,6], Taylor Chin [1,6], James A. Hay [1,6], Erik D. Surface[1], Pablo M. De Salazar [1], Caroline O. Buckee [1], Alexander Watts[2], Kamran Khan[2,3,4], Ryan Sherbo[2], Ada W. C. Yan[5], Michael J. Mina[1], Marc Lipsitch [1,7] & Rene Niehus [1,7✉]

Early in the COVID-19 pandemic, predictions of international outbreaks were largely based on imported cases from Wuhan, China, potentially missing imports from other cities. We provide a method, combining daily COVID-19 prevalence and flight passenger volume, to estimate importations from 18 Chinese cities to 43 international destinations, including 26 in Africa. Global case importations from China in early January came primarily from Wuhan, but the inferred source shifted to other cities in mid-February, especially for importations to African destinations. We estimate that 10.4 (6.2 – 27.1) COVID-19 cases were imported to these African destinations, which exhibited marked variation in their magnitude and main sources of importation. We estimate that 90% of imported cases arrived between 17 January and 7 February, prior to the first case detections. Our results highlight the dynamic role of source locations, which can help focus surveillance and response efforts.

[1] Center for Communicable Disease Dynamics, Department of Epidemiology, Harvard T.H. Chan School of Public Health, Harvard University, Boston, MA, USA. [2] BlueDot, Toronto, ON, Canada. [3] Li Ka Shing Knowledge Institute, St. Michael's Hospital, Toronto, ON, Canada. [4] Department of Medicine, Division of Infectious Diseases, University of Toronto, Toronto, ON, Canada. [5] Section of Immunology of Infection, Department of Infectious Disease, Imperial College London, London, UK. [6] These authors contributed equally: Tigist F. Menkir, Taylor Chin, James A. Hay. [7] These authors jointly supervised this work: Marc Lipsitch, Rene Niehus. ✉email: rniehus@hsph.harvard.edu

In late December 2019, a new severe acute respiratory syndrome coronavirus 2 (SARS-CoV-2) was identified, in Wuhan, Hubei Province, China[1]. Rigorous measures to curtail the spread of SARS-CoV-2, the causative virus of the COVID-19 disease, including travel restrictions and school and workplace closures, have largely controlled the outbreak in mainland China[2–5]. However, international exportation of COVID-19 cases before the outbreak was contained in mainland China ignited global spread of COVID-19[6], which has now become a pandemic. As of 31 August 2020, over 25 million confirmed cases of COVID-19 have been registered worldwide, with 90,000 detected in mainland China[7].

As the majority of cases in the early phase of the pandemic were reported in Wuhan, early COVID-19 case definitions and clinical guidelines required individuals suspected of infection to have had a recent travel history from Wuhan[8]. Similarly, models predicting internationally imported cases from China and local outbreaks in North America, Europe and Asia have largely relied on flight passenger numbers from Wuhan[6,9]. Based on those models, the risk for outbreaks in several African countries was estimated to be relatively low[9]. However, a significant number of COVID-19 cases were introduced to other travel hubs in China before travel restrictions were instituted on 23 January 2020[10,11]. This suggests that there may have been undetected imported cases globally, and thus, a potentially elevated risk of importation to African countries.

As of 31 August 2020, >1 million confirmed cases have been reported in all 54 African locations as defined by the United Nations[7]. Coverage of COVID-19 diagnostic and control interventions in many of those locations is expanding yet still limited, and many of these nations will continue to struggle to meet increased demand[12–14]. Furthermore, in the absence of reliable estimates of prevalence, it is extremely difficult to assess the current and future burden of COVID-19 in many of these locations. In light of these challenges, accurately estimating the timing and number of initial importations is crucial to inform models of outbreak dynamics for those locations.

Previous work has combined travel data with incidence estimates to estimate the risk of importation from all Chinese provinces, excluding Hubei, to all African countries[15]. That analysis, however, used historical flight data from January 2019, which is unlikely to reflect 2020 travel trends, given that the Lunar New Year occurred comparatively early (in January) in 2020 and unprecedented travel restrictions were in place starting late January. In addition, the analysis did not take into account the uncertainty in reporting rates, the delay between infection and case report, and the time-varying prevalence of infected individuals in China.

Here, we have developed a modeling framework to synthesize available travel and COVID-19 prevalence data to explore geographical and time-varying trends of international case importation. We first estimated daily flight passenger numbers from 18 major cities in China to 43 international destinations from December 2019 to February 2020. Importantly, we used current data on air travel that takes into account increased travel in early January due to the Lunar New Year holiday and reduced travel in late January due to travel restrictions. We then estimated daily COVID-19 prevalence in each Chinese province, taking into account delays between infection, symptom onset and confirmation, and differences in ascertainment rates between Hubei and other Chinese provinces. Finally, we combined air travel and prevalence estimates to give the daily number of internationally imported cases from mainland China to 43 international locations, including 26 destinations in Africa and globally representative locations on every continent. A number of assumptions are required for our predictions (e.g., relative ascertainment rates between Hubei and other provinces and the duration that each infection contributes to prevalence). Therefore, we present results from a range of scenarios capturing uncertainty in our key assumptions, with point estimates from our selected best-estimate scenario. Our findings reveal a shift in the importance of Wuhan versus other cities as a source for COVID-19 importations from China over the course of the early period of the pandemic, and a generally higher importance of non-Wuhan cities for case importations to African locations. Further, our model estimates the time window in which early local outbreaks may have been initiated from China and signals heterogeneities in the extent and the exact source of introduced cases in the African locations.

## Results

**Estimating the number of airplane passengers from China to global destinations.** We estimated the daily air-travel volume, defined as the daily number of passengers on direct and indirect air-travel routes, from 18 different Chinese cities to 43 international destinations. We estimated this volume within our focal time period, between 1 December 2019, (the approximate date of initiation of the pandemic in China[16]), and 29 February 2020, as the most recent date for which we have flight information (Supplementary Fig. 1). The 18 Chinese origin cities have been previously identified as high-risk cities for importation of COVID-19 from Wuhan[11]. Our 43 destinations included (1) ten high-surveillance locations that have high-surveillance capacity index (discussed previously[6]) and high air-travel connectivity to Wuhan, (2) 26 destinations in Africa (African locations) with high air-travel connectivity to the 18 Chinese origin cities and finally (3) seven additional locations that together with a subset of (1) and (2) yield 16 globally representative locations, which receive worldwide the highest air-travel volume from China and also represent every inhabited continent. Air-travel volume on each day of our focal period for each origin-destination pair was calculated using monthly air-travel volume (number of flight passengers), which we apportioned into days using daily flight departures (number of departing flights, see Methods).

The resulting trends in daily air-travel volume reflect the timing of the Lunar New Year holiday in 2020, which occurred earlier than in recent years, and also the impact of widespread travel restrictions in late January 2020 (Supplementary Fig. 1). Air-travel volume increased in January during the Chunyun period—the 40-day period surrounding Lunar New Year in which people typically reunite with their families that began on 10 January 2020—relative to the same time period in 2019 (Supplementary Fig. 1). The sharp decline in air-travel volume after 23 January 2020 is a result of the travel restrictions and flight cancellations that occurred starting late January (Supplementary Fig. 1).

**Estimating daily prevalence of COVID-19 in 18 Chinese cities.** Next, we estimated a daily prevalence indicator for all Chinese cities considered in our analysis. We defined this prevalence indicator as a measure that is linearly proportional to the actual (unobserved) prevalence of infected cases that are able to travel. See Methods for a detailed description. In brief, we first estimated the province-level daily incidence of onsets for detected infections by shifting confirmed case count curves using the delay between infection and symptoms (incubation period) and the delay in reporting (Fig. 1A and Supplementary Fig. 2). We then adjusted our estimates of incidence in Hubei to account for lower ascertainment of cases in Hubei relative to other provinces in China, in order to yield a measure of relative incidence in each province. Next, we converted our estimates of relative incidence to estimates of relative prevalent cases, by assuming that each newly

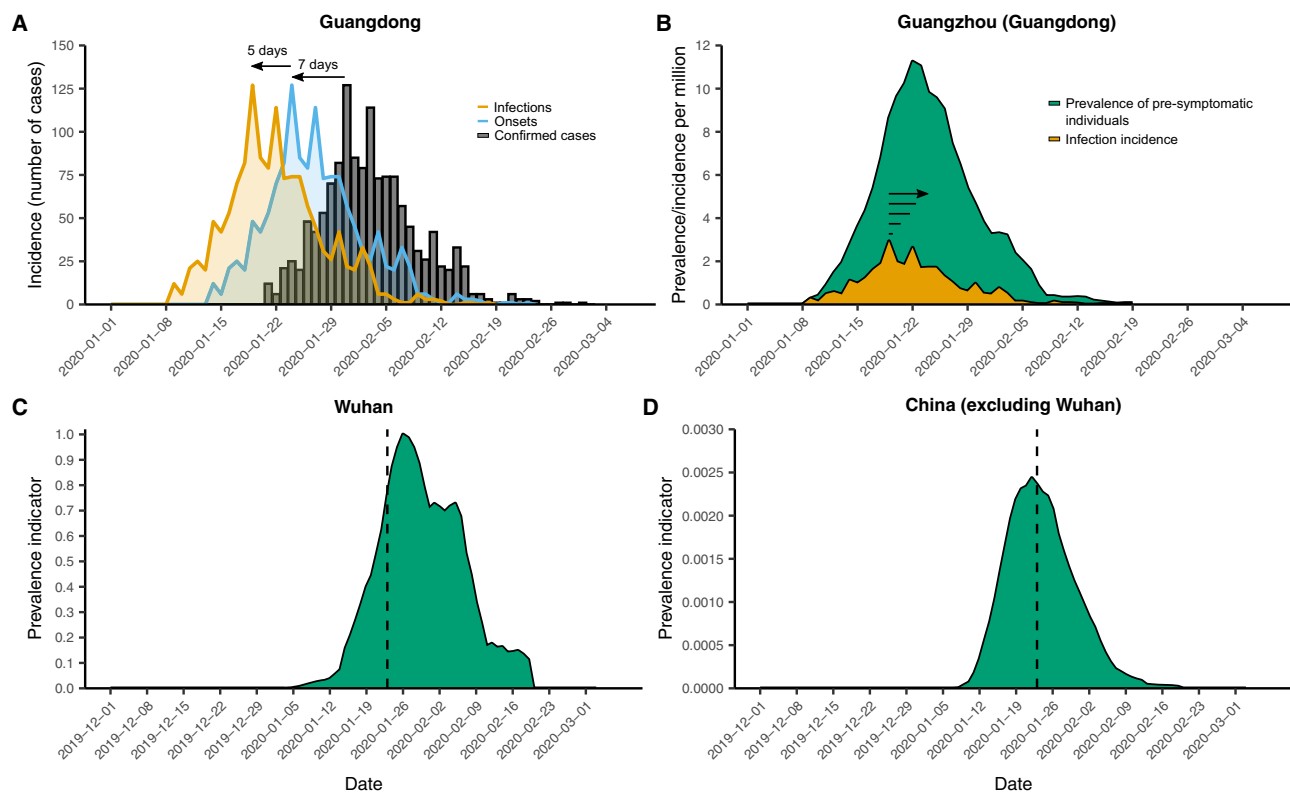

**Fig. 1 Number of reported COVID-19 cases and computed prevalence indicators. A** Estimation of symptom onsets (blue) and infections (orange) from observed confirmed case counts (gray bars) using Guangdong province as an example. Horizontal axis shows time given as calendar days. Confirmed cases were shifted back by 7 days (the mean confirmation delay) to estimate symptom onset incidence, then further by 5 days (the median incubation period) to estimate infection incidence of those cases. **B** Conversion of province-level infection incidence to city-level pre-symptomatic prevalence in Guangzhou (capital city of Guangdong province). Each infected individual (orange) was assumed to contribute towards pre-symptomatic prevalence (green) for 5 days (the median incubation period). Incidence and prevalence are shown per-capita and before adjusting for within-China differences in ascertainment rates. **C** and **D** Prevalence indicator (green area) for Wuhan and averaged for non-Wuhan cities (note ~300-fold smaller values on the y-axis in **D**). For curves of this indicator, only relative comparisons are meaningful, and are thus scaled relative to the maximum value in Wuhan. Vertical dashed line shows 23 January 2020, the date of lockdown in Wuhan.

infected case contributed to province-level prevalence for a number of days before they were no longer included in the travel-relevant pool of infected individuals (e.g., due to isolation upon symptom onset) (Fig. 1B). The prevalence indicator represents a relative measure of prevalence as it reflects true prevalent cases scaled to account for greater under-ascertainment in Hubei compared to that of other provinces. Finally, we allocated those province-level prevalence indicators to each city included in our analysis and standardized by population size to compute the corresponding city-level prevalence indicators (Supplementary Fig. 3).

In our best-estimate scenario (Scenario 1), we assumed that (1) each new infected case may travel for up to 5 days before showing symptoms (i.e., the median incubation period[17]), that (2) case ascertainment is 5.1 times as high outside of Hubei as it is in Hubei (see Methods), and that (3) per-province prevalence is allocated to cities proportional to their share of confirmed cases in the province by the end of the focal period. Under these assumptions, we found that the prevalence indicator peaked in all Chinese cities between 19 and 26 January 2020 (Fig. 1C, D and Supplementary Fig. 3). Other than Wuhan, Chinese cities with the highest average prevalence indicators were Nanchang, Jiaxing, and Chongqing, whereas Nanjing, Dongguan, and Tianjin had the lowest.

To account for the considerable uncertainty in the true time-varying SARS-CoV-2 infection prevalence in Wuhan and elsewhere in China during the initial outbreak, we varied each

of the three key assumptions to assess their impact on our subsequent analyses of case importations, leading to eight additional scenarios (Supplementary Table 1 and Supplementary Fig. 3). A key alternative scenario used estimated incidence curves from a previous analysis that accounted for time-varying ascertainment rates in China due to changing case definitions (Scenario 2)[18]. In contrast to our best-estimate scenario, which shifts and inflates only observed case counts, this alternative estimate suggested substantial undocumented incidence throughout December and early January (Supplementary Fig. 3). Note that the prevalence indicators from Scenario 2 provide a measure of absolute rather than relative prevalence, in contrast to all other scenarios. Under Scenario 2, prevalence indicators peaked on 20 January 2020 in all locations. For a detailed description of the other model scenarios see Methods section.

**Predicting exported case counts to African countries**. We combined our estimates of the daily COVID-19 prevalence indicator in China with the daily air-travel volume between China and a given set of destinations to obtain an indicator that is linearly proportional to the daily flight volume of infected travelers for each origin-destination pair, which we term the "force of importation". To translate the force of importation into the expected actual number of imported cases to a destination, we extended on the approach outlined by Salazar et al.[9] and fit a Poisson regression model to the cumulative number of imported and detected cases (with identified source location Wuhan) in our

high-surveillance locations from 1 December 2019 to 23 January 2020, when the lockdown was instituted in the city. For this we adjusted each destination country's detected case count by an estimate of underreporting[6] and focused only on imported cases from Wuhan, as we assume that a majority of imported cases during this period of the pandemic originated from Wuhan. We use the Poisson model fit for high-surveillance locations to obtain an estimate of the factor by which our estimated forces of importation must be scaled to predict the number of imported cases. Here, uncertainty around the nine scenarios for the pre-valence indicator described in section "Estimating daily pre-valence of COVID-19 in 18 Chinese cities" (model uncertainty) far exceeded uncertainty from fitting the Poisson regression (statistical uncertainty), so we chose to only present variation in our results across the nine scenarios. We bound our point esti-mates (i.e., mean predictions from Scenario 1) with the range of mean predictions across all nine model scenarios.

Our model estimates in our African destinations a total of 10.4 (6.2–27.1) imported COVID-19 cases, with all imports predicted on days prior to the first case detections in each location. The highest numbers of imports are expected for South Africa (3.0; 1.2–5.4) and Algeria (1.4; 0.2–1.7), followed by Kenya (1.2; 0.6–2.4), Zambia (1.1; 0.2–1.7), and Egypt (1.0; 0.6–5.1). Among the 26 African locations considered, we estimate the lowest imported case counts in Equatorial Guinea and Mauritania (both 0.02; 0–0.1) (Fig. 2). Figure 3A depicts the weekly predicted number of imported cases in each of the 26 African countries over time, highlighting the top five countries for which we predict the highest force of importation. All countries exhibit similar

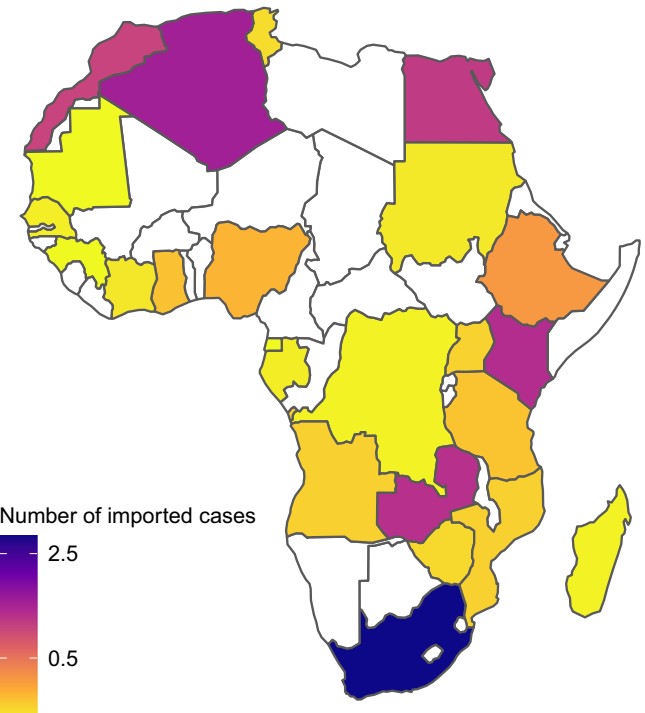

**Fig. 2 Predicted case importations for African locations.** Map of 24 African locations used in the analysis (two locations not shown are the island nations Seychelles and Mauritius, but see Supplementary Table 2 for predictions for all locations) with countries shaded by the magnitude of predicted imported cases from 18 Chinese cities during our focal time period (1 December 2019 to 29 February 2020) under our best-estimate scenario (Scenario 1). Light yellow indicates small values, dark blue color indicates high values. Countries shaded in white are locations for which we do not have data for prediction.

trajectories of predicted imported cases over the focal time period, with the exceptions of Algeria and Zambia, for which our estimates suggest a slightly delayed initial increase in cases relative to the other top-ranking locations. 90% of all imported cases were estimated to be imported between 17 January and 7 February. To contextualize the predicted numbers for African locations, we compared them to the other global locations included in our analysis. We find that the African countries with the highest predictions (South Africa, Algeria, Kenya) are expected to have importation numbers comparable to our locations from South America (Brazil, Argentina, Chile). For detailed predictions for all countries of this analysis see Supplementary Fig. 4.

**Contribution of different locations in China to globally imported cases.** We estimated the relative importance of Wuhan versus other cities as a source for international case importations from China and explored how these relative roles changed over time. To do so, we computed the daily forces of importation (as described above in section "Predicting exported case counts to African countries") to our globally representative locations with two different sources: (1) Wuhan, and (2) the 17 other Chinese cities. We found that early on in the pandemic, the majority of imported cases originated in Wuhan (100%; 70%–100% in the week of 1 January 2020), but this proportion then changed rapidly. The outlying lowest estimate of 70% corresponds to Scenario 2, which predicts substantial prevalence in all provinces in December and early January and higher prevalence in non-Wuhan cities relative to Wuhan compared to the other scenarios. In early January and late February, the proportion of globally imported cases sourced in Wuhan begins to change non-mono-tonically, dropping precipitously to 0% in the week of 19 Feb-ruary (across scenarios, Supplementary Fig. 5). This indicates a dramatic change in the contribution to imported cases from Wuhan relative to the rest of China. Note that before the con-tribution of Wuhan is drastically reduced due to the lockdown, one can observe a slight increase in the proportion of cases attributed to Wuhan in late January and early February that can be explained by the rapidly increasing prevalence during a second peak of disease activity (Supplementary Fig. 2) during this period.

The contribution of different source populations is expected to further vary across destinations. For the African destinations in our analysis, Wuhan contributed a majority of importations early in the epidemic (100%; 42%–100% in the week of 1 January 2020, where 42% corresponds to Scenario 2), subsequently declining to 0% (across scenarios) in mid-February (the week of 19 February). Figure 3C further illustrates the variability in the 18 Chinese cities' contribution to predicted imports over the study period across the 26 African destinations. In addition to Wuhan, Beijing, Guangzhou, and Shanghai consistently rank among the top cities in terms of their relative contribution to imported cases to each of the African countries included in our analysis.

**Sensitivity analyses of estimated prevalence indicators.** Scenario 2 demonstrates how our estimates change under very different assumed epidemic dynamics, as it predicts substantial case pre-valence in all locations as early as December 2019 (Fig. 3B). Scenario 2 predicts that 6.2 cases were exported to the 26 African locations during the focal period, which is the lowest of all of the tested scenarios. Notably, only 2.0 of these were predicted to have come from Wuhan as a result of higher predicted prevalence in non-Wuhan cities relative to Wuhan (Supplementary Fig. 3). Again, all of these importations were predicted to have occurred prior to the first identification of cases in these countries. The top five African countries with the most imported cases were South

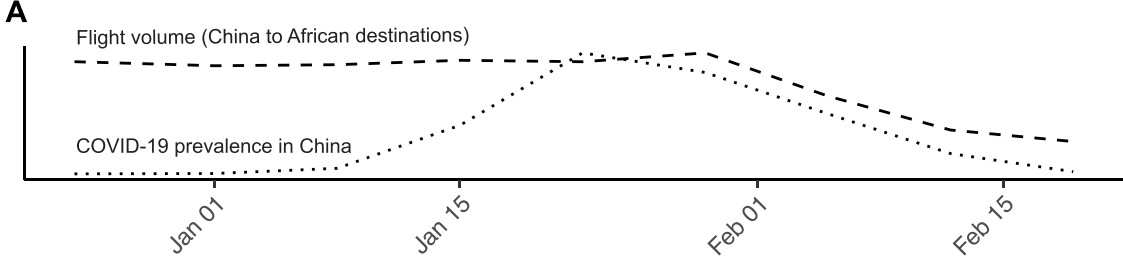

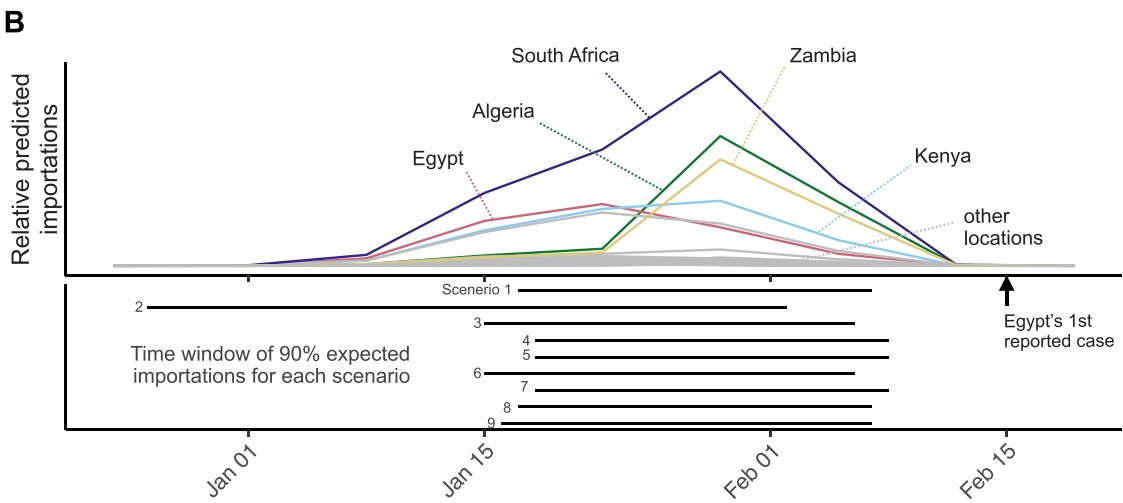

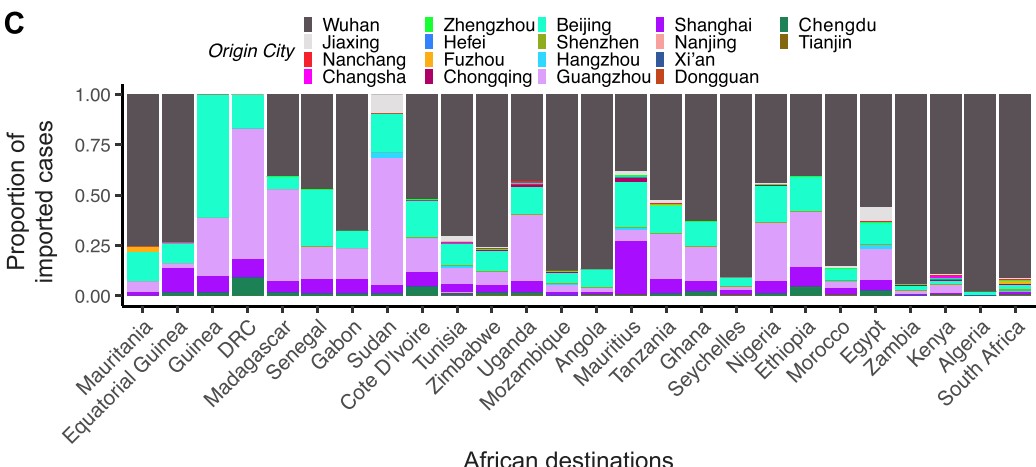

**Fig. 3 Predicted COVID-19 importations to African locations over time and by source city. A** Daily prevalence indicator (dotted line) summed across all 18 origin Chinese cities (including Wuhan), from our best-estimate scenario, and weekly flight volume from those cities to African destination countries (dashed line) over time. Prevalence peaks on 19 January 2020, while total flight volume peaks on 29 January 2020. **B** Colored and gray curves show weekly predicted number of cases for different destinations in Africa under Scenario 1. The first case on the African continent was reported in Egypt, on 15 February 2020[7]. Our nine model scenarios predict very consistent time windows in which 90% of imported cases are predicted to have arrived, barring Scenario 2 (shown as solid horizontal bars; bottom panel). **C** Rank of 18 Chinese cities by fraction of all predicted imported COVID-19 cases in each of the 26 countries in Africa included in our analysis, under Scenario 1. Countries are ranked from left to right by the total number of imported cases from 1 December 2019 to 29 February 2020. Origin cities are ranked from the bottom to top of each column by maximum estimated prevalence.

Africa (1.2), Egypt (1.2), Morocco (0.6), Kenya (0.6) and Ethiopia (0.4). The lowest number of imported cases were again predicted for Equatorial Guinea and Mauritania. Zambia and Algeria, which were within the top five in Scenario 1, were predicted to have the

7th and 8th highest number of imports (each with 0.2 importations) according to Scenario 2. These changes in ranking highlight the importance of taking into account the combined effects of flight and prevalence trends when assessing importation risk.

Together wih Scenario 2, our eight prevalence scenarios tested the robustness of our estimates to uncertainty in three key parameters: (1) ascertainment rates in Hubei relative to non-Hubei provinces; (2) the time that an individual contributed towards the prevalence indicator post-infection; (3) the allocation of province-level case reports to the cities considered here. For the best-estimate scenario (Scenario 1), we assumed that case under-ascertainment was five times as high in Hubei than outside of Hubei. Varying the ascertainment rate ratio for Hubei to non-Hubei by one order of magnitude in either direction (0.5:1 and 50:1) in Scenarios 6 and 7 changed our estimates for the absolute number of imported cases but did not change our inferred temporal patterns (Supplementary Table 1). When we assumed that under-ascertainment was 50 times greater in Hubei than outside of Hubei (lower proportion of true cases ascertained in Hubei), the absolute number of predicted importations from cities outside of Hubei dropped to 0.2 cases. Conversely, when assuming that under-ascertainment was twice as high outside of Hubei relative to Hubei, the absolute number of predicted imports from cities outside of Hubei increased substantially to 18.6 cases. These changes are tied to our calibration of the force of importation: if true prevalence were higher in Hubei but the observed number of imports in high-surveillance locations is unchanged, then the relatively lower prevalence in non-Hubei locations results in fewer exported cases.

In contrast, varying our assumption for the duration that an infected individual remained infected and eligible for international travel from 5 days to 2 and 7 days (Scenarios 3 and 4) did not meaningfully change our results (Supplementary Table 2). Although a shorter prevalence duration decreases the magnitude of the prevalence indicator, it has the same impact across Wuhan and non-Wuhan cities and is compensated by the calibration procedure described in section "Predicting exported case counts to African countries", so has negligible impact on the results. We also accounted for the observation that a substantial fraction of cases reported outside of Wuhan originated from Hubei and were thus only eligible to travel from a non-Wuhan city for the latter stage of their infection, again noting minimal impact on our results (Scenario 5). Next, rather than assuming that only cases reported in a city were able to travel internationally from that city's airport, we instead either apportioned all reported cases in a province to each city, or to cities equal to their fractional share of that province's population (Scenario 8 and 9, see Methods). These scenarios did change the relative contribution of different Chinese cities to total imports due to changes in per-capita prevalence, though the top contributing cities remained largely unchanged: Under Scenario 1, the top five contributing cities to importations in Africa were Wuhan, Guangzhou, Beijing, Shanghai and Jiaxing respectively. Under Scenario 8, these were Wuhan, Jiaxing, Guangzhou, Beijing and Chengdu, and under Scenario 9 these were Wuhan, Beijing, Guangzhou, Shanghai and Chengdu. However, overall trends in the number of imported cases over time in the African destination countries remained relatively unchanged for Scenarios 8 and 9, as did the total number of imports (11.7 and 12.5, respectively).

## Discussion

In this study we aimed to make predictions about internationally imported COVID-19 cases from all of China. Our analysis differs from that of previous studies in three fundamental ways: (1) instead of estimating risk of importation[15], our model predicted actual number of imported cases, and importantly did so for countries on the African continent; (2) instead of accounting only for travelers from Wuhan[9], we accounted for travelers from all major Chinese airports as a potential source population; and (3)

we incorporated current air-travel data from December 2019 to February 2020 and back-calculated prevalence from reported COVID-19 cases in China.

These methods in combination enabled us to predict daily time-varying case importations over the first three months of the pandemic. Our model predicted that until 29 February 2020 10.4 (range: 6.2–27.1) COVID-19 cases from all of China could have been imported to the 26 African destinations included here. Importantly, our model provided a relatively precise time frame for those importations. It predicted that the majority (90%) of case importations in these locations occurred between 17 January and 7 February. All predicted cases would have been undetected: the first African country to confirm a COVID-19 case was Egypt, which confirmed its first case 14 February 2020[19], followed by Algeria and Nigeria, with their first confirmed cases on 25 February and 28 February 2020, respectively[20]. If our predictions are accurate, then undetected imported cases could have already occurred a month before the first COVID-19 cases were confirmed, updating our understanding of the possible timing of when local transmission may have started in those locations. In the absence of strong surveillance systems, estimates of the current and future prevalence rely on dynamic transmission models, with a key unknown being the seeding time of the outbreak[21,22]. Our results provide such estimates of when index cases may have arrived in different African countries.

We further observed pronounced differences in the number of expected cases imported to the different African destinations. The highest numbers are expected for South Africa, Algeria, Kenya, Zambia, and Egypt, the lowest numbers for Mauritania and Equatorial Guinea. These heterogeneities in predicted imports and the relatively early lockdown of borders in several African countries may at least in part explain early differences in the scale of outbreaks in these countries, with some experiencing outbreaks of considerable magnitude and others seemingly spared. Moreover, variation across African countries in the timing of the peaks of predicted importations (Fig. 3B) may be due to differences in the dynamics of air-travel volume, or due to differences in the proportion of passengers flying from certain Chinese cities.

On 31 March 2020, a month after the end of our prediction period, South Africa (1326 cases), Egypt (609 cases), and Algeria (584 cases) were the countries in Africa with the most reported cumulative cases and likely location transmission[23], while Equatorial Guinea (14 cases) and Mauritania (5 cases) reported relatively few cases[24], with unlikely local transmission[23]. Thus, our predictions of the top five locations with the most expected importations generally align with the observed case counts one month later. Notably, however, two of the top five locations—Kenya and Zambia—had relatively low case counts on 31 March 2020[23]. We expect that for different locations and as time passes, our predictions may diverge due to a number of factors beyond importations from China that likely had a significant influence on the observed epidemics in those locations, including sourcing from newly emerging epicenters at the time (for example Europe), differences in response measures, reporting and testing capacity across countries, as well as travel between African countries.

A strength of our method framework is that it rests on a relatively small number of assumptions. For example, it does not rely on estimates of actual case prevalence in China; instead, case counts in high-surveillance locations are used to relate relative force of importation to absolute numbers of cases. It does, however, necessitate assumptions about relative ascertainment rates in Hubei compared to other provinces. The calculation for our main scenario assumes similar death ascertainment rates, infection fatality rates and proportions of asymptomatics between Hubei and outside Hubei. Various factors are likely to violate these assumptions, including differential strain on health care and

surveillance systems and different subpopulations being affected by disease spread (for example, local spread versus importation by travelers). Thus, to reflect the uncertainty in estimates of this ratio, our analysis included additional scenarios that assumed a time-varying ratio, as well as a tenfold lower or tenfold higher ratio (Scenario 2, 6, and 7 respectively, see Supplementary Table 1). Similarly, we included scenarios to express uncertainty in how many days infected cases may travel and are detectable (Scenario 3 and 4, Supplementary Table 1), and in how detected cases affect prevalence at airport hubs (Scenario 8 and 9, Supplementary Table 1).

Additionally, our method assumed a fixed value of reporting delays over time, although delays were in fact estimated to decrease from late January onwards[5]. If reporting delays indeed declined over time, such that cases reported from late January[5] onwards actually became incident cases at a later date than that which we inferred, then this could influence the tail end of the trajectory of predicted imported cases. However, this is unlikely to influence the initial trends in predicted imported cases, the timing of the first introduced cases, and the overall magnitude of imported cases. Finally, our estimates using daily flight departure data may only approximately reflect the number of passengers flying each day, as more departing flights might not directly correspond to increased flight passenger numbers and vice versa. This approximation will be increasingly violated in times when demand–and thus airplane capacity–drastically changes, as was probably the case in early 2020. Second, sparsity of recorded flight departures required us to smooth over particularly sparse time periods. However, we expect that these assumptions will have a relatively muted impact on the timing of predicted importations.

It is important to note that the magnitude of total predicted imports to the African locations under our best-guess scenario was relatively modest, suggesting limited importations from China, which is a direct consequence of the scale of the case data on which we train our model. Specifically, due to low reported case counts in some of our included validation countries, our estimated forces of importation from Wuhan align well with imported cases in these locations, and thus do not require a substantial adjustment. As such, our estimates of the forces of importation for all other cities to our African locations must only be marginally inflated to predict imported case counts in these destinations. Furthermore, while we expect the presence of asymptomatics among airline passengers to result in increased predicted imports, by amplifying our estimated forces of importation to these locations by an additional factor, we do not expect this to influence the observed shape in the imported case curves over time, if the proportion of asymptomatics among cases is constant over time. Finally, our scenarios capture different epidemic trajectories, in terms of the initial emergence and rate of increase in prevalence, and thus reflect a wide range of possible dynamics of imported cases in the African destination countries. These scenarios all indicate an early timing of initial imported cases, where the scenario assuming a more measured and early increase in prevalence (Scenario 2) places the introduction of the earliest case far before the remaining scenarios.

Recent work has estimated that, globally, 2.8 cases imported from Wuhan may have remained undetected for every 1 detected case due to limited surveillance capacity[6]. Our predictions also highlight potential underestimation; when considering all of China as a source of importation–and assuming equivalent sensitivity for detecting cases from Wuhan and other Chinese cities —67% (65%–82%) of all cases imported globally may have been undetected (see Methods). Wuhan was identified early as the major source population based on its high COVID-19 prevalence. Here, we show the importance of spill-over to locations in the rest of mainland China, and that possible source populations strongly depend on actual travel volume. Some locations may have relatively low SARS-CoV-2 prevalence, but greater connectivity to a given destination country would still result in a high overall number of imported infections. Our model predicts for the early pandemic that between 70%-100% of globally imported cases came from Wuhan in the week of 1 January 2020, with the rest originating from other Chinese cities. We find this proportion dropped to 0% in the week of 19 February. This sheds light on how profoundly source populations can change over time under the effect of lockdowns as well as a rapidly spreading virus. In addition, the relative importance of a source population also depends on the destination of imported cases. We found that for the African locations the share of cases exported from Wuhan in the week of 1 January 2020 was slightly lower than that of all destinations, ranging from 42%–100% in the week of 1 January 2020, but similarly declining to 0% in mid-February.

Our findings highlight the importance of a more nuanced understanding of likely sources of case importation for predictive modeling. In sum, this framework allows for routinely quantifying recently imported, and potentially overlooked cases and assessing the principal sources of importation to prioritize in surveillance efforts among travelers, particularly during the later stages of this pandemic when travel restrictions are eased. Going forward, countries wishing to identify likely sources of case importations may benefit from combining ongoing travel volume and prevalence data as we have done here, allowing for more nuanced policy decisions than those based on global trends. Our approach further elucidates the potential time window for the first imported cases, which may or may not have been successful in seeding local transmission. This approach may help to initialize models aimed at anticipating future trends in COVID-19 transmission in diverse locations. Importantly, these methods can be adjusted to incorporate prevalence estimates and flight data from any number of origin and destination locations with minimal data requirements: reported case data in the source and calibration locations, daily flight departure data and monthly flight passenger totals. The tools we propose here are particularly useful for locations facing significant surveillance constraints and potential resource limitations in managing ongoing response efforts, and thereby work to address enduring gaps in infectious disease monitoring and preparedness.

## Methods

**Estimating number of airplane passengers from 18 Chinese cities to international destinations**. We used data on the number of passengers and flight departures from 1 December 2019 to 29 February 2020 from 18 Chinese origin cities to 43 international destinations, as described below. Historical air-travel volume data are likely not representative of the pandemic time period for two reasons: (1) Lunar New Year was earlier than in preceding years (25 January 2020), and (2) large-scale travel bans and flight cancellations took place in late January 2020.

We defined three main categories of locations as international destinations in our analyses (1) ten international locations with high-surveillance capacity and high air-travel connectivity to Wuhan used for model calibration; (2) 26 African countries as destinations used for model prediction; and (3) 16 locations that represent the top three destinations (two for Oceania and North America) from each inhabited continent (Oceania, Europe, Asia, Africa, North America, South America) in terms of travel volume from China during our focal time period. Those 16 locations include locations from (1) and (2) and an additional seven locations. We used these locations to estimate the ratio of cases imported internationally from other Chinese origin cities compared to cases imported internationally from Wuhan. For all flight data, in addition to Wuhan, we included as origin locations ($N_0 = 17$) the 17 Chinese cities that were previously identified by Lai et al.[11] as high-risk cities for importation of COVID-19 from Wuhan and, therefore, likely sources of imported cases internationally: Beijing, Shanghai, Guangzhou, Zhengzhou, Tianjin, Hangzhou, Jiaxing, Changsha, Nanjing, Nanchang, Shenzhen, Chongqing, Chengdu, Hefei, Fuzhou, Xi'an, and Donngguan.

For destinations outside of China, we considered a selection of locations with both high air-travel connectivity to Wuhan and high-surveillance capacity (henceforth called high-surveillance locations) for model validation. We assessed surveillance capacity using the Global Health Security (GHS) Index, in particular its components assessing early detection and reporting epidemics of potential international concern, published in 2019[6]. We thus selected locations with the highest connectivity to Wuhan as estimated by Lai et al.[11], and within the top 5% percentile of the GHS index rank. We additionally included Singapore as it has demonstrated a strong capacity to identify, trace and document COVID-19 cases[9,25], despite having a relatively low GHS index.

In total, we used $N_D = 43$ total destination locations: (1) the ten high-surveillance locations of Singapore, US, Australia, Canada, South Korea, UK, Netherlands, Sweden, Germany, and Spain for model validation; (2) 26 African countries for prediction, representing the 26 top destination cities in Africa in terms of air-travel volume from 18 high-risk cities in mainland China (Nigeria, Ghana, Algeria, Côte D'Ivoire, Ethiopia, Egypt, Guinea, Morocco, Tanzania, Senegal, South Africa, Uganda, DRC, Zimbabwe, Sudan, Angola, Gabon, Zambia, Mozambique, Mauritania, Mauritius, Kenya, Seychelles, Madagascar, Tunisia, Equatorial Guinea); and (3) 16 global main destinations from China (New Zealand, Australia, United Kingdom, Germany, Russia, Japan, Thailand, South Korea, United States, Canada, Brazil, Argentina, Chile, Egypt, Ethiopia, South Africa).

We used data from the International Air Transport Association (IATA)[26] on the monthly number of confirmed passengers on flights (direct and indirect) for each of the $N$ origin-destination pairs, henceforth referred to as air-travel volume, from December 2019 to February 2020. In addition to air-travel volume, we used data from Cirium[27] on the number of daily departed (and landed) direct passenger flights for each of the $N$ origin-destination pairs, henceforth referred to as "flight departures," for the period December 2019 to February 2020.

We combined the Cirium data on daily flight departures with the IATA monthly air-travel volume data to estimate daily air-travel volume out of cities in China to our destinations of interest from 1 December 2019 to 29 February 2020. For all origins except for Wuhan, we distributed the monthly air-travel volume in month $m$, $V_{i,j,m}$, into daily air-travel volume for each day $d$, using the proportion of daily flight departures out of the total number of daily flights in the corresponding month.

$$v'_{d,i,j} = V_{i,j,m} \frac{\sum_{j=1}^{N_D} f_{d,i,j,m}}{\sum_{j=1}^{N_D} \sum_d^{d_{max}} f_{d,i,j,m}}$$

where $v'_{d,i,j}$ is the imputed air-travel volume from origin $i$ to destination $j$ on day $d$, $f_{d,i,j,m}$ is the number of landed flight departures on day $d$ from origin $i$ to destination $j$ in month $m$, $N_D$ is the number of destination locations. When distributing the monthly air-travel volume, we used the proportion of daily flight departures for each origin $i$ summed over all destinations $j$ instead of calculating this for each origin-destination connection due to sparse data limitations.

The same approach could not be applied for Wuhan due to sparsity in the number of direct landed flight departures from Wuhan in the Cirium data after 23 January 2020. For Wuhan, to distribute monthly air-travel volume, $V_{i,j,m}$, into daily air-travel volume for each day, we instead fit a smoothing spline to the daily number of landed flight departures and used its predictions as inputs in the above equation. We used the forward-chaining time-series cross-validation procedure, applying an accumulative rolling training window of 7 days, in order to estimate the smoothing parameter for the spline[28,29].

Since residents of the cities Shenzhen and Dongguan share Shenzhen Bao'an International Airport as the closest international airport, we divided the air-travel volume equally between these two cities. Similarly, the air-travel volume for Hangzhou Xiaoshan International Airport airport is divided equally for the cities Jiaxing and Hangzhou. Importantly, Cirium data only documents the daily number of direct flights, but we use this data source to distribute a total 3-month volume into daily volume. We therefore make the assumption that the variation over time in the number of direct flights reflects the variation in the number of direct and indirect flights.

**Estimating daily prevalence of COVID-19 in 18 Chinese cities**. We used data on the number of confirmed COVID-19 cases reported by China CDC per day by province[31,32]. To estimate daily incidence, we backwards shifted the time-series of confirmed cases by a mean reporting delay of 7 days[5] to yield the total number of symptom onsets per day. We shifted this onset incidence curve backwards again by the median incubation period of 5 days[17] to yield the total number of infection onsets who had not yet developed symptoms. The mean reporting delay is estimated using line list data summarized in Zhang et al.[5], which provides information on the number of days between which an individual develops symptoms and is reported as a case.

We then scaled infection onset curves by province by our computed ascertainment rate ratio (ARR) as of 29 February 2020, to generate estimates of relative incidence in each province. The ARR represents the ascertainment rate of symptomatic cases in all other provinces relative to that of Hubei (e.g., an ARR of 5:1 implies that the probability a true (symptomatic) case was reported was five times as high in non-Hubei provinces relative to Hubei). We define the

ascertainment rate among cases, rather than total infections, as we calibrate our forces of importation on reported cases. We derived this ascertainment rate ratio using data on confirmed cases, confirmed deaths, an estimate of the infection fatality ratio (IFR), and an estimate of the proportion of asymptomatic infections. In particular, we use cumulative confirmed deaths by Chinese province 13 days following 29 February (13 March), which captures the expected delay between case confirmation and confirmation of death[30]. The quotient of the cumulative deaths and the infection fatality ratio (IFR) yield total infections (*total infections = cumulative deaths/IFR*). We use an IFR of 0.66% following Verity et al.[31], but note that the IFR cancels out so that other values for the IFR (see for example Emery et al.[32] or Mizomoto et al.[33]) would not change our resulting ARR. The ascertainment rate ratio (ARR) is then defined as follows:

$$ARR = \frac{\text{cumulative confirmed cases(NH)}}{\text{total infections(NH)}*\text{prop}_{symptomatic}} \bigg/ \frac{\text{cumulative confirmed cases(H)}}{\text{total infections(H)}*\text{prop}_{symptomatic}} = AR(NH)/AR(H),$$

(1)

where $\text{prop}_{symptomatic}$ denotes the proportion of infections that becomes symptomatic. Both for non-Hubei (here denoted with NH) and for Hubei (H), we divide the cumulative daily reported counts of cases by an estimate of how many cases could have been detected—given as the total infections times the proportion of symptomatic infections. This provides an estimate for both the non-Hubei and Hubei ascertainment rate. Taking the ratio of the two rates yields the relative ascertainment rate of non-Hubei provinces relative to Hubei. For the proportion of infections that are symptomatic we use 26%[32], but any other value for this proportion (see for example Van Vinh Chau et al.[34] or Ward et al.[35]) would generate the same ARR.

As an alternative to our back-shifted incidence curves, we also used estimated onset incidence curves from a previous Bayesian analysis of SARS-CoV-2 onset incidence in China[18] (Scenario 2). Tsang et al.[18] make the crucial point that case ascertainment driven by changes in case definition over time would have a significant impact on the inferred dynamics of the epidemic. The authors used a Bayesian analysis assuming exponential growth (with a different rate before and after 23 January 2020) to infer the number of case onsets that would have been observed had a later, broader COVID-19 case definition been used throughout the outbreak. For Wuhan, we used the posterior mean estimated onset incidence curve (Fig. 3 of Tsang et al.[18]), which estimates onset incidence in Wuhan if the COVID-19 case definition as of 4 February 2020 had been applied throughout. For non-Wuhan cities, we took the posterior mean estimated onset incidence curve for China excluding Wuhan divided among each Chinese province proportional to the number of confirmed cases in the province from China CDC data[36,37]. Finally, we back-shifted these onset curves by the median incubation period as above.

For all scenarios, we estimated the travel-related prevalence indicator (corresponding to absolute prevalence in the case of Scenario 2) each day by summing over individuals who were infected on a given day and individuals who were infected in previous days and have not yet developed symptoms (Supplementary Table 1). To convert our estimates of the province-level prevalence indicator to per-capita prevalence in each Chinese city, we attributed prevalent infections in each province (absolute numbers) to each city proportional to their share of province-wide confirmed cases as of 13 March (same date as used for the death counts above). This attribution assumes that case ascertainment rates are comparable between our city hubs and other parts of the provinces, and also that infected cases are as likely to travel as the general population. The true distribution of cases at the city-level likely deviates from both of these assumptions, which may shift true case attribution in either direction. Thus, as a sensitivity analysis, Scenario 8 assumes that province-level prevalence is allocated entirely to the cities considered in our analysis, or divided equally across our cities when they share a province. In so doing, we assume that individuals in a province are equally likely to go to the specific airports in our analysis. For example, we assumed that 100% of cases in Jiangxi province were in Nanchang with a population of around 2.4 million, whereas we assumed that one third of cases in Guangdong were in Guangzhou, Dongguan and Shenzhen respectively. As a further sensitivity analysis, Scenario 9 instead assumed that prevalent cases were attributed to cities proportional to the city's share of its province's population. For example, the city of Hefei accounts for 5.23% of the population of Anhui and was, therefore, assumed to account for 5.23% of all infections in Anhui. Scenario 8 and 9 encompass the two extremes of how reported cases affect the traveling populations in airport hubs and are likely to bracket the true scenario. As a final step, we divided the total number of cases by the population size of that city to generate per-capita prevalence estimates.

**Estimating number of imported cases to international destinations**

*Model training: associating flight volume of infected passengers from Wuhan with observed number of Wuhan-origin cases in validation set locations.* We first fit a model to the number of imported COVID-19 cases from Wuhan observed in the high-surveillance locations to determine the relationship between prevalence indicator, air-travel volume and imported case counts. We used this model fit to make predictions using data from Wuhan and the remaining cities in China. The number of observed cumulative cases imported from Wuhan to destination $j$ prior to the 23 January is denoted as $y_j$. Further, $y_j^* = 2.5y_j$ denotes the number of

cases that each destination location $j$, excluding Singapore, could have detected with a surveillance capacity of Singapore[6] (for Singapore $y^*_{j=\text{Singapore}} = y_{j=\text{Singapore}}$). Following the analysis by Niehus et al.[6], 2.5 represents the ratio of Singapore's capacity to identify imported cases to that of other high-surveillance countries. We assumed that across the high-surveillance locations (U.S., Australia, Canada, South Korea, UK, Netherlands, Sweden, Germany, Spain, and Singapore), this number follows a Poisson distribution, as follows:

$$y_j \sim \text{Poisson}\left(\frac{\alpha C_{w,j}}{2.5}\right)$$

$$C_{w,j} = \sum_t \text{prev}_{w,d} * v_{d,w,j}, \tag{2}$$

where $C_{w,j}$ represents the *force of importation* from Wuhan to each destination $j$, which is calculated as the product of the COVID-19 prevalence indicator in Wuhan ($\text{prev}_{w,d}$) and volume of passengers from Wuhan to destination $j$ ($v_{d,w,j}$) on day $d$, summed over all days in the pre-lockdown pandemic period, and $\alpha$ represents the scaling factor relating the *force of importation* to scaled reported cases in the *high-surveillance locations*, $y^*_j$. We fit this model using the *glm* function in R (version 3.6.1)[38].

*Model application: predicting imported case counts to subset of African destinations.* We defined the pre-lockdown pandemic period (referring to the lockdown of Wuhan) and our focal pandemic period, which we considered to be from 1 December 2019 (approximate date of seeding in Wuhan[15]) to 23 January 2020 and from 1 December 2019 to 29 February 2020, respectively.

The force of importation of COVID-19 from all selected cities in China to destination $j$ in Africa was computed as:

$$C_j = \alpha \sum_i^{N_0+1} \sum_t \text{prev}_{i,d} * v_{d,i,j}, \tag{3}$$

where $\text{prev}_{i,d}$ is the prevalence indicator of COVID-19 in Chinese city $i$ at time $d$, and $v_{d,i,j}$ is the total volume of passengers across flights from each origin city $i$ to each destination $j$ on day $d$. The product of daily passenger volume ($v_{d,i,j}$) and the COVID-19 prevalence indicator in city $i$ ($\text{prev}_{i,d}$) was summed over all days of the focal pandemic time period, and over all $N_0$ Chinese cities and Wuhan. We used this force of importation to make predictions for all 26 African locations under each of the nine scenarios for the $N_0$ Chinese cities.

*Proportion of all imported cases from Wuhan and from other cities in China.* To estimate the proportion of all expected cases imported from Wuhan and from other Chinese cities, we computed the force of importation from Wuhan ($C_{w,j}$, as defined above), the force of importation from all 18 origin cities ($C_j$, as defined above, where $j$ now represents all destination locations), and the force of importation from the 17 Chinese cities, excluding Wuhan, as follows:

$$C_{\overline{w},j} = \sum_i^{N_0} \sum_t \text{prev}_{i,t} * v_{t,i,j}, \tag{4}$$

where the product of daily passenger volume ($v_{t,i,j}$) and the COVID-19 prevalence indicator in city $i$ ($\text{prev}_{i,t}$) under Scenario 1 is summed over all days of the focal pandemic period, and over the 17 Chinese cities (excluding Wuhan). The quotient of $C_{w,j}$ and $C_j$ gives the proportion of all imported cases to all destinations j from Wuhan and the quotient of $C_{\overline{w},j}$ and $C_j$ gives the proportion of all imported cases to all destinations j from all 17 other origin cities (here, denoted as NW), under each scenario, according to:

$$\text{Prop}(W) = \frac{C_{w,j}}{C_j} \text{ and } \text{Prop(NW)} = \frac{C_{\overline{w},j}}{C_j} = 1 - \text{Prop}(W). \tag{5}$$

We computed this proportion for two different sets of locations ($j \in \{$African locations$\}$ and $j \in \{$all locations$\}$), for all scenarios. In order to estimate the proportion of all cases imported globally that may have been undetected, we assumed based on recent work that 2.8 cases imported from Wuhan may have been undetected for every one detected case due to limited surveillance capacity[6]. We then scale our estimated proportion of imported cases that originated from Wuhan by this factor (i.e., $\frac{2.8}{\text{Prop}(W)}$) to estimate the total number of possible importations from all the origin cities in China. Finally, the proportion of global importations that may have been undetected was calculated as $1 - \frac{1}{\frac{2.8}{\text{prop}(W)}}$.

**Reporting summary.** Further information on research design is available in the Nature Research Reporting Summary linked to this article.

## Data availability

International Air Transport Association (IATA) data was provided by BlueDot, Toronto, Canada (https://bluedot.global). Data on departed flights was provided by Cirium (https://www.cirium.com). This data is not publicly available as raw data. Transformed flight data used for this work is freely available online (Menkir et al.[39]). COVID-19 case counts are from publicly available datasets: WHO reports (https://apps.who.int/iris/bitstream/handle/10665/331425/SITREP_COVID-19_WHOAFRO_20200311-eng.pdf) and European CDC (https://opendata.ecdc.europa.eu/covid19/casedistribution/csv), and MIDAS reports (https://midasnetwork.us/covid-19/). The derived COVID-19 prevalence data and importation risk generated in the analysis is available online (Menkir et al.[39]).

## Code availability

No data collection was needed. We used readily available data on COVID-19 epidemiology, and we were provided data on air-travel (see Data section). Analysis and visualization was performed using R (version 4.0.1) and Rstudio (1.3.959). All code and data are available online (Menkir et al.[39]).

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

## Acknowledgements

We thank Mauricio Santillana, Lee Kennedy-Shaffer, Rebecca Kahn, Christine Tedijanto, Justin Lessler, Kylie Ainslie, and Charlie Whittaker for their valuable input and feedback. M.L., T.F.M., T.C., J.A.H., M.J.M., and R.N. were supported by Award Number U54GM088558 from the US National Institute Of General Medical Sciences. P.M.D. was supported by the Fellowship Foundation Ramon Areces. C.O.B. was supported by a NIGMS Maximizing Investigator's Research Award (MIRA) R35GM124715-02. M.J.M. was supported by an NIH DP5 grant DP5OD028145. The study was supported in part by the Morris-Singer Fund for the Center for Communicable Disease Dynamics. The funding bodies of this study had no role in the study design, data analysis and interpretation, or writing of the manuscript. The content is solely the responsibility of the authors and does not necessarily represent the official views of the National Institute Of General Medical Sciences or the National Institutes of Health.

## Author contributions

T.F.M., T.C., J.A.H., P.M.D.S., M.L., and R.N. designed the study. T.F.M., T.C., J.A.H., A.W.C.Y., and R.N. performed the analyses. T.F.M., T.C., J.A.H., P.M.D.S., and R.N. wrote the manuscript. All authors (T.F.M., T.C., J.A.H., E.D.S., P.M.D.S., C.O.B., A.W., K.K., R.S., A.W.C.Y., M.M., M.L., R.N.) reviewed and approved the final manuscript.

## Competing interests

M.L. has received consulting fees from Merck. All other authors declare no competing interests.
