## [Peer Review File · Nature Communications]

REVIEWER COMMENTS

Reviewer #1 (Remarks to the Author):

As the authors mentioned, major findings have been published in Journals such as Lancet Infectious Diseases. The findings in this paper is minimal and should not be published in Nature

In the manuscript titled “Estimating internationally imported cases during the early COVID-19 pandemic”, the authors combine highly resolved air travel data during the early phase of the pandemic, with epidemiological data on imported SARS-CoV-2 cases in “high-surveillance” countries, to estimate the opportunities of importing SARS-CoV-2 infections from 18 international transportation hubs of China in African countries. Based on the analysis presented in the manuscript, the authors emphasize that cities other than Wuhan could play an equally important role in seeding the epidemic globally, and in particular, in Africa. The topic of the manuscript is timely and of interest to a board audience, as effective surveillance and intervention measures at the early stage of an emerging epidemic are the most cost-effective way in preventing potential pandemic events. The manuscript is well written, and the methodologies are clearly presented. However, I find a few assumptions made by the authors may not be realistic and shall be addressed through revision, among other minor comments. Specifically:

- From line 161-169, the authors clearly outlined the assumptions made for the best-estimate scenario. However, for assumption 2) “case ascertainment is twice as high outside of Hubei as it is in Hubei” may not accurately reflect the reality. This assumption on case ascertainment rate rely on data from a very nice and carefully conducted study by Verity et. al. However, the study was conducted back in March and rely on relatively outdated data in February. During that period, a majority of the outcomes (deaths in particular) of reported SARS-CoV-2 infections are unknown. As the initial epidemic waves has subsided in China and it has been clear now that the case fatality rate outside Hubei is much lower when compare that in Hubei (although may present spatial variations), in directly reflecting a higher (likely significantly more than a factor of two in many locations) case ascertainment rate outside Hubei. Empirically, the epidemic scenarios are very different inside and outside Hubei: within Hubei, the community transmission was widespread and with limited testing capacity in the early days, only severe cases are being detected; outside Hubei in general, people who had recent travel history to Hubei were being screened and tested, and very extensive contact-tracing efforts were in place to quarantine population with high-risk exposure (both travel-related and locally). As a result, community transmission was quite limited and did not trigger large scale outbreaks outside Hubei province and case ascertainment rate is likely much higher outside Hubei (likely comparable to situations in “high surveillance countries”). I recommend the authors consider updating case ascertainment rate with relatively recent evidence (for example, use reported death to backward infer total infections using current estimate on IFR, then calculate reporting rate with the estimated denominator of total infections).
- For assumption 3) “all cases reported in a province came from the cities/city included in our analysis”. This is unfortunately a quite biased assumption for provincial level administrative areas with more than one city within the province. In general the cities considered in the study only account for a fraction of the total cases reported in the province, and in many cases were not even the cities/city reported most cases in the province (noting that Wuhan is a well-established industrial hub in China with substantial migrant workers and the province of Hubei serves simultaneously as both labor exportation and importation to other provinces, thus a significant fraction of people leaving Wuhan during Chinese New Year represent population that works in Wuhan, but

came from cities with relatively lower socioeconomic status in other provinces. Spatially resolved data of SARS-CoV-2 cases and deaths at city level can be found at:

- MIDAS network: https://github.com/midas-network/COVID-19/tree/master/data/cases/china/cumulative_cases_DXY/
- service provided by Chinese social-media obtained from local health officials: <https://news.qq.com/zt2020/page/feiyun.htm#/?nojump=1>

And can be verified at provincial/local health commission websites.

- To put case importations in Africa from China at a global perspective, how does total importations in the African continent when compared to Europe, Asia, and Americas? I recommend the authors provide a simple histogram of total importations by continent.
- The authors presented 9 scenarios in total in the main text. In my opinion, this is a bit distracting and difficult to follow. I recommend the authors reduce the number of scenarios presented in the main text (3-4 would be ideal) and present the rest of the scenario analysis in the supplementary material.
- A minor point: line 139, “Chunyun” is not a holiday, it’s a word describing the migration of population during the Chinese New Year, when people typical reunite with their families throughout the country. You may simply say the “Chunyun” period.
- Line 439-440, missing values in “% (%-%).

Response to reviewers' comments (reviewer comments shown in full in italics):

Reviewer #1:

As the authors mentioned, major findings have been published in Journals such as Lancet Infectious Diseases. The findings in this paper is minimal and should not be published in Nature.

We respectfully disagree, and think that the findings in this paper form substantial contributions to our understanding of the ongoing COVID-19 pandemic. Our results suggest that a substantial amount of early COVID-19 importation likely evaded surveillance because international case definitions for COVID-19 required a recent travel history from Wuhan, and we also quantify this amount. Further, we highlight how source populations change rapidly due to quickly evolving local epidemics and changing travel behaviour, something that—to our knowledge—has not been studied in this way. Finally, we provide a time window for the earliest potential seeding opportunities for local epidemics in these locations, estimating that 90% of imported cases arrived between 17 January and 7 February, prior to the first case detections. Our method combines robust COVID-19 prevalence estimates with the most recent data on airplane passenger numbers from China and we think that this is an important and generalisable modeling framework that allows estimation of the number of infections imported across borders accounting for evolving contribution of different source populations. All of these questions are of importance at all stages of the COVID-19 pandemic and of epidemic outbreaks in general.

Reviewer #2:

In the manuscript titled “Estimating internationally imported cases during the early COVID-19 pandemic”, the authors combine highly resolved air travel data during the early phase of the pandemic, with epidemiological data on imported SARS-CoV-2 cases in “high-surveillance” countries, to estimate the opportunities of importing SARS-CoV-2 infections from 18 international transportation hubs of China in African countries. Based on the analysis presented in the manuscript, the authors emphasize that cities other than Wuhan could play an equally important role in seeding the epidemic globally, and in particular, in Africa. The topic of the manuscript is timely and of interest to a broad audience, as effective surveillance and intervention measures at the early stage of an emerging epidemic are the most cost-effective way in preventing potential pandemic events. The manuscript is well written, and the methodologies are clearly presented.

Many thanks for this clear and enthusiastic summary of our work.

However, I find a few assumptions made by the authors may not be realistic and shall be addressed through revision, among other minor comments.

Thank you for these insightful and specific suggestions. We have changed the assumptions going into our main model scenario in line with these suggestions, and have updated our

quantitative estimates throughout the main text. These new assumptions have reduced the overall number of predicted imported cases and suggest a decreased but still significant role for non-Wuhan cities relative to our initial assumptions. Our findings regarding the timing of importations and heterogeneities across African destinations are largely unchanged.

Specifically:

- *From line 161-169, the authors clearly outlined the assumptions made for the best estimate scenario. However, for assumption 2) “case ascertainment is twice as high outside of Hubei as it is in Hubei” may not accurately reflect the reality. This assumption on case ascertainment rate rely on data from a very nice and carefully conducted study by Verity et. al. However, the study was conducted back in March and rely on relatively outdated data in February. During that period, a majority of the outcomes (deaths in particular) of reported SARS-CoV-2 infections are unknown. As the initial epidemic waves has subsided in China and it has been clear now that the case fatality rate outside Hubei is much lower when compare that in Hubei (although may present spatial variations), indirectly reflecting a higher (likely significantly more than a factor of two in many locations) case ascertainment rate outside Hubei. Empirically, the epidemic scenarios are very different inside and outside Hubei: within Hubei, the community transmission was widespread and with limited testing capacity in the early days, only severe cases are being detected; outside Hubei in general, people who had recent travel history to Hubei were being screened and tested, and very extensive contact-tracing efforts were in place to quarantine population with high-risk exposure (both travel-related and locally). As a result, community transmission was quite limited and did not trigger large scale outbreaks outside Hubei province and case ascertainment rate is likely much higher outside Hubei (likely comparable to situations in “high surveillance countries”). I recommend the authors consider updating case ascertainment rate with relatively recent evidence (for example, use reported death to backward infer total infections using current estimate on IFR, then calculate reporting rate with the estimated denominator of total infections).*

Many thanks for this constructive comment. We agree that many factors (including the ones mentioned above) speak for a larger than two-fold difference in ascertainment rates between Wuhan and outside. This was in fact the reason that we had included a scenario with a much more extreme ascertainment ratio of 20:1 (scenario 7). Following the above suggestion, we have now used more recently available data to update our best estimate scenario.

We have used reported COVID-19 death counts in China matching our focal time period, sourced from the MIDAS *github* repository suggested below, to obtain a new estimate for the ascertainment rate in Wuhan relative to non-Wuhan cities. We derive this estimate using the number of deaths and IFR as suggested, but note that this estimate is independent of the IFR and asymptomatic fraction since they cancel in the calculation. However, we have included those rates to make the calculation easier for the reader to follow (see *Methods* section *Estimating daily prevalence of COVID-19 in 18 Chinese cities*). We have used this new estimate for our main, best estimate scenario (see *Results* section *Estimating daily prevalence of COVID-19 in 18 Chinese cities*), and we have also added a discussion addressing the

assumption going into this new scenario (see *Results* section *Estimating daily prevalence of COVID-19 in 18 Chinese cities* and see also *Discussion* section). In addition to this scenario we performed sensitivity analyses using either a 10-fold higher ascertainment ratio (new Scenario 6) or 10-fold lower ratio (new Scenario 7) to demonstrate how our results change if our ascertainment rate assumptions are misspecified. We discuss the impact that varying this model assumption has on our results (see *Results* section *Sensitivity analyses of estimated prevalence indicators*).

For assumption 3) “all cases reported in a province came from the cities/city included in our analysis”. This is unfortunately a quite biased assumption for provincial level administrative areas with more than one city within the province. In general the cities considered in the study only account for a fraction of the total cases reported in the province, and in many cases were not even the cities/city reported most cases in the province (noting that Wuhan is a well-established industrial hub in China with substantial migrant workers and the province of Hubei serves simultaneously as both labor exportation and importation to other provinces, thus a significant fraction of people leaving Wuhan during Chinese New Year represent population that works in Wuhan, but came from cities with relatively lower socioeconomic status in other provinces. Spatially resolved data of SARS-CoV-2 cases and deaths at city level can be found at:

o MIDAS network:

https://github.com/midas-network/COVID19/tree/master/data/cases/china/cumulative_cases_DX Y/ o service provided by Chinese social-media obtained from local health officials:

<https://news.qq.com/zt2020/page/feiyang.htm#/?nojump=1> And can be verified at provincial/local health commission websites.

Thank you for this comment. We fully agree that attributing all cases from a province to a city is a biased scenario (likely attributing too many cases to the cities). For that reason we had included the other extreme of that assumption: province cases are attributed to cities proportionally to the city population relative to the province population (likely attributing too few cases to the cities). Arguably, the truth lies somewhere in between these two extremes. The suggestion to use city-level case data is a great approach to providing an estimate between these two extremes.

We have made use of the above suggested data source (provided by the MIDAS network) to estimate the per-province share of cases of our cities using city-level cumulative confirmed cases. While death counts are expected to be more reliably ascertained than cases, cumulative deaths are relatively sparse on the city level and thus make it difficult to reliably estimate city shares of the outbreak. The resulting way of allocating prevalent cases to cities has now become our new best estimate scenario (see *Results* section *Estimating daily prevalence of COVID-19 in 18 Chinese cities*). We have further added a discussion to address the assumptions for this new scenario (see *Results* section *Estimating daily prevalence of COVID-19 in 18 Chinese cities*), and we include the two extreme scenarios described above (all province cases attributed to our cities, or cases attributed according to relative city-population)

as alternative scenarios (new Scenario 8 and 9), and we discuss the effect of this key assumption on our results (see *Results* section *Sensitivity analyses of estimated prevalence indicators*).

To put case importations in Africa from China at a global perspective, how does total importations in the African continent when compared to Europe, Asia, and Americas? I recommend the authors provide a simple histogram of total importations by continent.

Thank you for this suggestion. We have now added a new figure (new Supplementary Figure S4) showing predicted case importations to all 43 locations included in our analysis. Colour coding by geographical region here allows the reader to easily compare across continents. See below the new figure for reference:

New Supplementary Figure S4. Plot showing the predicted number of COVID-19 case importations (using a log-scaled vertical axis) from China to all countries included in our analysis. Points show predictions of the best-estimate model, lines indicate ranges across all model scenarios. Countries are sorted from highest to lowest number predicted cases and are colored by continent. Note that countries from each continent were selected based on having high average flight volume from China.

The authors presented 9 scenarios in total in the main text. In my opinion, this is a bit distracting and difficult to follow. I recommend the authors reduce the number of scenarios presented in the main text (3-4 would be ideal) and present the rest of the scenario analysis in the supplementary material.

We agree that less model scenarios would help the narrative of the main text. Our aim was to be transparent about our key assumptions, but we appreciate that presenting all of the results together is distracting. We have therefore now adapted the methodological suggestions above to form a new best-estimate scenario that we describe in full detail in the main text. For example, Figure 1 now shows only the prevalence indicator estimates for the best-estimate scenario, and the corresponding information for the remaining 8 scenarios has been moved to Supplementary Figure S3. To make the alternative model scenarios clearer, we have grouped

them by the model assumption being varied (Supplementary Table S1). These are i) the duration that a case contributes towards prevalence, ii) the ascertainment rate ratio, and iii) the allocation of province-level cases to cities as described above. We now present only the main Scenario 1 in the main text, discuss the impact of these assumptions on our key results (see *Result* section *Sensitivity analyses of estimated prevalence indicators*) and provide results from the other 8 scenarios in the Supplementary Material.

A minor point: line 139, "Chunyun" is not a holiday, it's a word describing the migration of population during the Chinese New Year, when people typically reunite with their families throughout the country. You may simply say the "Chunyun" period.

Thank you for pointing this out. We have now changed the wording to Chunyun period, and adjusted our explanation of the period.

Line 439-440, missing values in "% (%-%).

Many thanks, we have filled in the missing value.

REVIEWERS' COMMENTS

Reviewer #2 (Remarks to the Author):

The authors have fully address my previous comments and now the manuscript is significantly improved. I have no further comments and recommend manuscript for publication.